# Accuracy of Pulse Oximetry in the Presence of Fetal Hemoglobin—A Systematic Review

**DOI:** 10.3390/children8050361

**Published:** 2021-04-30

**Authors:** Ena Pritišanac, Berndt Urlesberger, Bernhard Schwaberger, Gerhard Pichler

**Affiliations:** 1Research Unit for Neonatal Micro- and Macrocirculation, Medical University of Graz, Auenbruggerplatz 34/II, 8036 Graz, Austria; ena.pritisanac@medunigraz.at (E.P.); berndt.urlesberger@medunigraz.at (B.U.); bernhard.schwaberger@medunigraz.at (B.S.); 2Division of Neonatology, Department of Pediatrics, University Hospital Graz, Auenbruggerplatz 30, 8036 Graz, Austria

**Keywords:** neonate, fetal hemoglobin, oxygen saturation monitoring, pulse oximetry

## Abstract

Continuous monitoring of arterial oxygen saturation by pulse oximetry (SpO2) is the main method to guide respiratory and oxygen support in neonates during postnatal stabilization and after admission to neonatal intensive care unit. The accuracy of these devices is therefore crucial. The presence of fetal hemoglobin (HbF) in neonatal blood might affect SpO2 readings. We performed a systematic qualitative review to investigate the impact of HbF on SpO2 accuracy in neonates. PubMed/Medline, Embase, Cumulative Index to Nursing & Allied Health database (CINAHL) and Cochrane library databases were searched from inception to January 2021 for human studies in the English language, which compared arterial oxygen saturations (SaO2) from neonatal blood with SpO2 readings and included HbF measurements in their reports. Ten observational studies were included. Eight studies reported SpO2-SaO2 bias that ranged from −3.6%, standard deviation (SD) 2.3%, to +4.2% (SD 2.4). However, it remains unclear to what extent this depends on HbF. Five studies showed that an increase in HbF changes the relation of partial oxygen pressure (paO2) to SpO2, which is physiologically explained by the leftward shift in oxygen dissociation curve. It is important to be aware of this shift when treating a neonate, especially for the lower SpO2 limits in preterm neonates to avoid undetected hypoxia.

## 1. Introduction

Continuous arterial oxygen saturation measured by pulse oximetry (SpO2) is the primary monitoring to guide respiratory and oxygen support in neonates during postnatal stabilization and after admission to a neonatal intensive care unit (NICU) [1,2]. The recent resuscitation guidelines recommend specific pre-ductal SpO2 targets during postnatal transition based on the 25th percentile of SpO2 values in healthy term neonates that required no medical interventions at birth (2 min 65%, 5 min 85%, 10 min 90%) [1,3].

Before the 1980s, transcutaneous oxygen tension measurement (tc-pO2) was a common monitoring method in the NICU. Because of the practical aspects (regular calibration and repositioning of the electrodes, skin irritations, underestimation of partial oxygen pressure (paO2) in older neonatal patients) pulse oximetry was introduced into neonatal care as a better and more convenient monitoring method [4,5,6,7].

Pulse oximetry measures SpO2 by illuminating the tissue and detecting changes in the absorption of oxygenated and deoxygenated blood hemoglobin at two wavelengths: 660 nm (red) and 940 nm (infrared). In order to establish the pulse oximeter’s measure of SpO2, the ratio of absorbance at these wavelengths is calculated and calibrated against direct measurements of arterial oxygen saturation from blood samples (SaO2). For this purpose, blood samples are taken from healthy adult volunteers under room air (normoxemia) and in artificially acquired hypoxic environments to achieve hypoxemia [8,9].

The difference (bias) between SpO2 and SaO2 reported in adults is 3–4%, with a tendency for overestimation of SpO2 in critically ill mechanically ventilated patients [10,11,12]. However, studies conducted in mechanically ventilated neonates and children reported an even greater bias, particularly at lower SpO2 values. For instance, in the largest conducted study in children, the median bias of SpO2 versus SaO2 was as high as 6% for a SpO2 range of 81% to 85% [13]. Moreover, within the saturation target range for preterm infants (89–95%), pulse oximetry exceeded the 4% error quality margin in the latest published study, which included 1908 neonates. SpO2 values were overestimated by an average of 2.9% with a standard deviation (SD) of 5.8% in this study [14].

The oxygen carrying capacity of blood depends primarily on the hemoglobin molecule. Fetal hemoglobin (HbF) is the main oxygen carrier during pregnancy. From the 20th week of gestation, HbF is gradually replaced by adult hemoglobin (HbA) and declines to its adult levels by approximately six months after birth [15,16]. HbF exhibits a significantly higher affinity for oxygen, which enables oxygen extraction from the blood of the mother to the fetus via the placenta at lower partial oxygen pressures and leads to the shift of the oxyhemoglobin dissociation curve (ODC) to the left (shown in Figure 1) [17,18].

The prenatal HbF expression and conversion to HbA is regulated by a set of evolutionarily conserved genes and is not affected by the birth event itself. HbF values at birth are therefore particularly high in very low birth-weight neonates (HbF > 90%) [19]. However, in term neonates, these values can vary considerably among individuals, as reported in the largest conducted study in more than 150,000 newborns (mean HbF 82%, range 5–100%) [20]. Higher HbF values were observed in newborns exposed to risk factors for maternal or fetal hypoxia and for sudden infant death syndrome (SIDS) [21]. Furthermore, higher HbF values were reported to reduce the incidence of retinopathy of prematurity (ROP) in at-risk preterms, suggesting that HbF could be a protective factor for oxygen-related tissue injury in preterm neonates [22].

HbF content in the blood is often expressed as a percentage of total hemoglobin or fraction of fetal hemoglobin (FHbF) and can be measured by several methods. These include the alkali denaturation method, electrophoresis, spectroscopy, and high-performance liquid chromatography, which is the most accurate method and the gold standard. The differentiation between fetal and adult hemoglobin in a sample is based on the existence of gamma-chain peaks, which are characteristic of HbF. The level of HbF can be determined by measuring the total chromatogram gamma-globin chain areas expressed as a percentage of total Hb [23].

However, because of its wide availability, visible absorption spectroscopy performed by a hemoximeter or a blood-gas analyzer is the most commonly used method in clinical studies [24,25]. The optical system of a hemoximeter is designed to measure the concentration of total hemoglobin, oxygen saturation, and fractions of oxyhemoglobin, carboxyhemoglobin, deoxyhemoglobin, methemoglobin, and HbF. HbF does not have the same visible absorption spectrum as HbA due to a slight variation in molecular structure [26]. If not taken into account, the presence of HbF in a sample will interfere with the results of oxygen saturation and the carboxyhemoglobin. Newer models of the hemoximeter (since 1992) use a linear relationship to adjust the SaO2 and oxyhemoglobin readings by the measured level of HbF [27].

Since the calibration curves of pulse oximeters use SaO2 measurements from the blood samples of healthy adults (with almost no HbF), the accuracy of SpO2 values in the presence of HbF is questionable. The aim of this review was, therefore, to summarize the studies which examined the effect of HbF on pulse oximetry monitoring in human neonates.

## 2. Materials and Methods

Articles were identified using the stepwise approach specified in the Preferred Reporting Items for Systematic Reviews and Meta-Analyses (PRISMA) statement [28].

### 2.1. Search Strategy

A systematic search of Pubmed/Medline, Embase, Cumulative Index to Nursing & Allied Health (CINAHL) and Cochrane library was performed from the date of inception of the databases to January 2021 to identify articles that concerned HbF and oxygen saturation monitoring by pulse oximetry in term and preterm neonates. Only human studies written in the English language were selected. Search terms included: newborn, neonate, preterm, term, infant, HbF, hemoglobin F, fetal hemoglobin, after birth, postnatal, oxygenation, arterial oxygen saturation, pulse oximetry, SaO2 and SpO2 (Appendix A). Studies on fetal hemoglobin addressing sickle cell anemia and thalassemia were excluded. Additional published reports were identified through a manual search of references in retrieved articles and in review articles. The search was last updated on 24 January 2021.

### 2.2. Study Selection

Identified articles were independently evaluated by two authors (E.P., G.P.) by reviewing the titles and abstracts. If an uncertainty remained regarding the eligibility for inclusion, the full text was reviewed. The two reviewers independently selected relevant abstracts, critically appraised the full texts of the selected articles, and assessed the methodological quality of the studies. Data were analyzed qualitatively. Extracted data included the characterization of study type, patient characteristics, methods, and results.

## 3. Results

Our initial search identified 2024 articles. After the removal of duplicates, 1822 articles were screened for inclusion. Exclusion criteria included absence of reliable HbF measurements or non-invasive oxygenation monitoring in term or preterm neonates (shown in Figure 2). Ten observational studies fulfilled the inclusion criteria [4,5,6,7,29,30,31,32,33,34]. No randomized controlled trial was identified. All studies performed measurements of HbF and non-invasive oxygen saturation monitoring by pulse oximetry at the upper and/or lower extremity in neonates in the first days and weeks after birth and determined blood oxygenation parameters. The study populations included preterm and term neonates with a range of gestational ages from 24 to 42 weeks of gestation. Studies are presented in Table 1 and Table 2 according to the HbF measurement method.

Five studies conducted before 1992 used alkali denaturation or electrophoresis (Table 1), whereas five studies initiated after 1992 used a hemoximeter for the HbF measurement. (Table 2) All studies compared the non-invasive SpO2 readings to invasively measured blood oxygenation parameters, most commonly SaO2, and included HbF in the analyses. 

One out of the five studies conducted before 1992 found a 2.8–3.6% underestimation in SpO2 readings in relation to higher HbF levels [4], two found no bias in SpO2 readings in relation to HbF [7,32], and two reported inconclusive results [5,33]. Out of the five studies conducted after 1992, one reported no SpO2-SaO2 bias in relation to HbF [6], three studies reported an overestimation of SpO2 with higher HbF but did not provide statistical evidence to support this statement [29,30,31], and one study reported a decrease in SaO2-SpO2 bias following transfusion of adult blood to the neonates and consequential HbF decline. It remains unclear whether this decrease can be attributed to HbF alone [34].

## 4. Discussion

To our knowledge, this is the first systematic review on the influence of HbF on SpO2 monitoring in human neonates. Based on the results of the majority of the included studies, a SpO2-SaO2 difference (bias) can be detected when the SpO2 (%) readings are compared to the direct measurements of SaO2 (%) or HbO2 (%) in neonatal blood. Reported mean SpO2-SaO2 bias ranged from −3.6% (SD 2.3) to +4.2% (SD 2.4) (Appendix A). Although there have been indications that the bias could be influenced by HbF, none of the included studies provided an adequate statistical analysis to prove this statement.

We included ten studies in our analysis and divided them in two groups according to the technical characteristics and HbF measurement methods (Table 1 and Table 2).

The five studies listed in Table 1 were conducted before the automatic correction of SaO2 for the presence of HbF by the hemoximeter (before 1992) [4,5,7,32,33]. Therefore, the corrections were performed retrospectively using a formula suggested by Cornellison et al. [35]. SpO2-SaO2 bias, which could be attributed to HbF, was detected in two studies [4,5]. The first study found a SpO2 underestimation of 2.8% to 3.6% for higher HbF values (FHbF > 50%) [4]. The second study reported a SpO2 overestimation for the lower HbF values (FHbF < 50%, SpO2-SaO2 bias +4.2% (SD 2.4)) and a decrease in SpO2-SaO2 bias for the higher HbF (for FHbF > 50%, SpO2-SaO2 bias +0.9% (SD 1.8)) [5]. Two of the five studies reported no significant effect of HbF on SpO2 accuracy [7,32]. Nevertheless, these two studies included patients with wide variations in HbF levels (FHbF 4–95%) and reported only the mean difference between SpO2 and SaO2 for all patients. The fifth study of the period before 1992 observed the effects of multiple factors (HbF, pH, pCO2, 2,3-DPG) on ODC in neonates and found that all of the parameters influenced ODC and therefore affected the corresponding SpO2 [33] (Figure 1). As the SpO2-SaO2 bias was not tested for HbF alone, the reported results are difficult to interpret.

Five studies conducted after 1992 used a hemoximeter for HbF measurements and adopted the automatic correction for SaO2 that accounts for the presence of HbF [6,29,30,31,34]. Out of these, one study found pulse oximeter saturations to be unaffected by HbF. It is important to mention that the 22 preterm neonates included in this study received multiple transfusions of adult blood which led to a rapid postnatal decline in HbF levels in the study population (FHbF 0–16% after 2 weeks). Moreover, the study reported an average SpO2-SaO2 bias from all of the acquired measurements irrespective of the HbF level at the time of the blood sampling [6].

The three larger studies by Shiao et al. reported primarily an HbF effect on SaO2 and HbO2 measurements from neonatal blood samples [29,30,31]. Although the authors mentioned the SpO2-SaO2 and SpO2-HbO2 bias, there was no statistical evidence that these could be attributed to HbF alone.

In their first study on 210 neonatal blood samples, the authors compared different measurement modes of the hemoximeter: the HbA-mode (adult mode) and the HbF-mode (fetal mode). They found that the blood saturation values were 4% to 7% higher using the HbA-mode as compared to the HbF mode (which assumed FHbF of 80%). The analyses with the HbA-mode overestimated both arterial and venous saturation from neonatal blood samples. Regarding the SpO2-SaO2 comparisons, a SpO2-SaO2 bias of −0.59% (2SD 5.93) for the HbF mode vs. −5.69% (2SD 5.96) for the HbA mode was reported. However, the bias was tested only for the arterial saturation range of 97.5% (SD 3.16) and there was no statistical analysis of HbF contribution to the SpO2-SaO2 bias. Based on these results, it is difficult to assess pulse oximeters’ accuracy for the different saturation ranges as well as the HbF contribution to the biases [29].

In their largest study, Shiao and Ou reported that the bias between SpO2 and HbO2 in arterial blood samples was as high as 2.5% (SD 3.1) for the arterial saturation range of 96.9% (SD 3.18). However, any influence of HbF is only reported on blood-derived oxygen saturation parameters and was not tested for SpO2-SaO2 bias. Nevertheless, the authors presented several ODC based on the paO2 and SaO2 of their samples and showed that the ODC in neonates was not only left-shifted but also steeper when compared to adults. For paO2 values between 50 and 75 mmHg (normoxemia), SpO2 ranged from 95% to 97% in neonates as compared to 85% to 94% in adults [31].

This narrow SpO2 range is based on the physiological characteristics of HbF. The study conducted on blood samples of extremely low birth weight neonates with very high HbF levels showed that a paO2 of 41 mmHg should be adequate to saturate 90% of HbF at a physiological pH. Therefore, the paO2 range of 45 to 60 mmHg could be defined as safe and preferable for this group of patients [36]. However, at paO2 of 50 mmHg, HbF is already 95% saturated. Consequently, further increase in paO2 leads to a minimal increase in saturation. (Figure 1) These observations further stress the importance of accurate SpO2 measurements and correct SpO2 targets to avoid undetected hypoxic or hyperoxic episodes.

Finally, the last included study, which investigated the effect of transfusion of adult blood and the consequential HbF decline on oxygenation parameters in neonates, found that there was a significant increase in paO2 after the transfusion (51 ± 8 mmHg vs 57 ± 7 mmHg, *p* < 0.001) with almost no changes in SpO2 (94 ± 2% vs 93 ± 1%, *p* = 0.4). This was achieved by an increase in FiO2 (>12%) applied to the infants to keep the SpO2 within the set goal [34]. However, it is not clear from this study whether the results reflect only the decrease in HbF or whether the changes in other parameters, such as pH or methemoglobin after the transfusion, might have influenced the described changes as well.

Based on the ten included studies, a SpO2-SaO2 bias can be detected by direct comparison of SpO2 readings to SaO2 in neonatal blood after the correction for HbF, but it is unclear to what extent this can be attributed to the HbF alone. An increase in HbF changes the relation of SpO2 to paO2, which is physiologically explained by the leftward shift in the ODC. It is important to be aware of this shift when treating a neonate, especially for the lower SpO2 limits in preterm neonates. Because of the fetal ODC form (Figure 1), a potential undetected hypoxia is particularly pronounced in the lower saturation ranges, i.e. for SpO2 < 90% where the curve is steep and becomes less detectable at its flat part (SpO2 > 95%). From this point of view, it can be assumed that there is only a low risk of undetected hyperoxemia when using an upper alarm limit of 95%. This was already shown in a study on three different pulse oximeters (Agilent Viridia, Masimo SET, Nellcor Oxismart), which detected hyperoxemia with 93–95% sensitivity for the upper alarm limit of 95% [37].

The question of optimal oxygen-saturation targeting for preterm neonates in order to avoid hypoxic and/or hyperoxic organ damage has been a subject of numerous, large, randomized controlled clinical trials [38,39,40,41,42,43,44]. Lower SpO2 target ranges (85–89%) have led to a decreased risk of retinopathy of prematurity but an increased risk of mortality [45]. If we took the ODC characteristics of HbF into account, the lower target ranges may have resulted in lower SaO2 values, as one would expect, and could have potentially resulted in more significant undetected hypoxemia in preterm infants. This may also have contributed to the reported increased rate of mortality and necrotizing enterocolitis in these patients. In addition, red blood cell transfusions, which are often required in preterm infants, lead to an increase in HbA relative to HbF, thus resulting in an ODC shift to the right. If the SpO2 target ranges are set higher, the ODC shift to the right after a transfusion may lead to hyperoxemia and increase the incidence of retinopathy of prematurity.

Finally, there are additional limitations of the included studies. The changes in the ODC positions (and consequently of SaO2) based on the differences in pH, temperature, and pCO2 (Figure 1) were not investigated in most of the studies. The studies also did not report the influence of oxygen supplementation on the SpO2-SaO2 bias. The largest study in neonates, which compared more than twenty-seven thousand SpO2 readings to SaO2 and paO2, however, reported a three-fold higher likelihood of SpO2 overestimation in infants treated with supplemental oxygen [14]. An additional explanation for the SpO2 differences in neonates and adults is that the sensors used in the calibration process of pulse oximeters have a different optical-path length in an adult compared to an infant, which may affect the accuracy of pulse oximeters in neonates [9,46]. As different measurement methods for HbF were used within the studies, this fact is a further limitation for the interpretation of the HbF levels and for the comparison of the studies. HPLC as a gold standard was only used in one study and when compared to the HbF measurements by a hemoximeter, a bias of 23% (SD 9) was detected [30].

## 5. Conclusions

In studies that compared non-invasive SpO2 monitoring by pulse oximetry to oxygen saturation measurements from blood samples in preterm and term infants and included HbF measurements in their reports, the majority found a SpO2-SaO2 bias, but it remains unclear whether this can be explained by the high fractions of HbF in neonatal blood alone. As hemoximeters today usually correct for the presence of HbF, SaO2 values of those devices likely reflect the paO2 of neonatal blood correctly. Based on the physiological characteristics of fetal ODC, there might be an influence of HbF on SpO2 readings, resulting mostly in an overestimation of SpO2 for the lower saturation ranges. Further prospective studies on a larger sample size are needed to support this statement. 

## Figures and Tables

**Figure 1 children-08-00361-f001:**
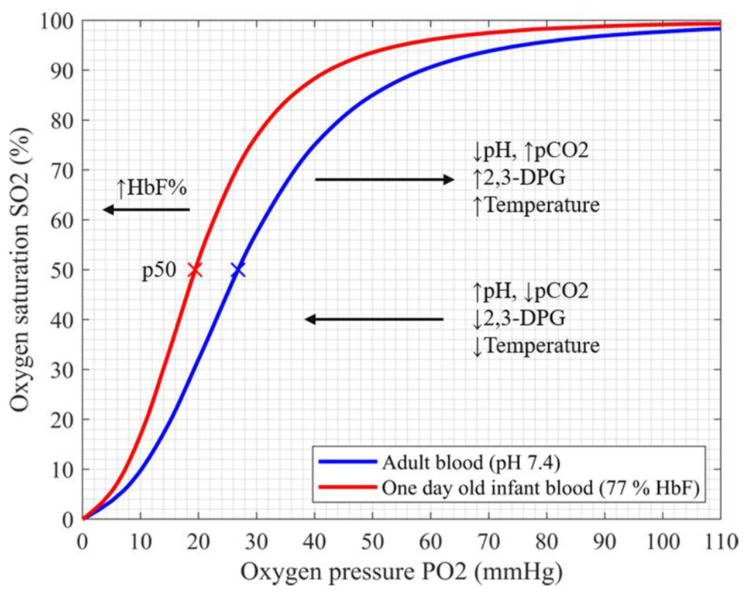
Oxyhemoglobin dissociation curve of fetal and adult hemoglobin shows the relationship between pO2 and SO2. For the saturation of 50%, the corresponding pO2 values (p50) are indicated (×). The factors that change the hemoglobin affinity for oxygen are indicated. HbF (red), fetal hemoglobin; pCO2, partial pressure of carbon dioxide; 2,3-DPG, diphosphoglycerate.

**Figure 2 children-08-00361-f002:**
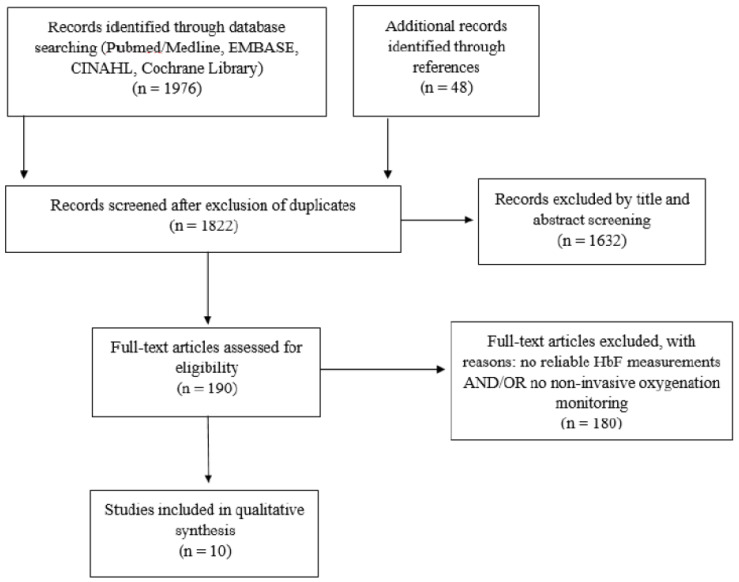
Study selection flow diagram.

**Table 1 children-08-00361-t001:** Studies before 1992 comparing SpO2 monitoring to invasively measured blood oxygenation parameters.

Ref	1st Author, Year	Number of Patients/ HbF Blood Samples	Blood Sample Type	HbF Measurement Method	Gestation Distribution (Weeks)	Time of Sample Collection and Non-Invasive Monitoring	Blood Oxygenation Parameters	Blood Gas Analyzer /Hemoximeter	Pulse Oximeter (Company Name)	Additional Bedside Oxygenation Monitoring Device (Company Name)	Relevant Results
[32].	Durand, 1986	75/140	Arterial	Alkali denaturation method	24–42	1–14 days + 30–153 days after birth	paO2, SaO2	Radiometer BMS3 Mark II / Co-oximeter IL 282	Nellcor N-100 (Hayward, CA, USA)	tc-pO2 Oxygen electrode (Novametrix, Wallingford, CT, USA)	HbF values of 4.3% to 95% did not influence the accuracy of pulse oximeter readings.
[7].	Ramanathan, 1987	68/132	Arterial	Alkali denaturation method	25–31	1–6 days + 20–80 days after birth	paO2, SaO2	Radiometer BMS3 Mark II / Co-oximeter IL 282	Nellcor N-100 (Hayward, CA, USA)	tc-pO2 Oxygen electrode (Novametrix, Wallingford, CT, USA)	HbF values of 4.3% to 92.2% did not influence the accuracy of pulse oximeter readings.
[33].	Wimberley, 1987	18/18	Arterial	Alkali denaturation method	25–34	Within 5 days after birth	paO2, SaO2	ABL300/ Hemoximeter OSM3	Ohmeda Biox 3700	tc-pO2 Radiometer TCM3	FHbF ranged from 44–97%. The variations in the levels of HbF, pH, pCO2 and 2,3-DPG resulted in a variable paO2-SaO2 relation.
[4].	Jennis, 1987	26/49	Arterial	Electrophoresis	24–40	1–49 days after birth	SaO2	Co-oximeter IL-282	Nellcor N-100 (Hayward, CA, USA)	NA	FHbF > 50% generated a 2.8% to 3.6% error (underestimation) in SpO2 reading.
[5].	Praud, 1989	71/52	Arterial	Electrophoresis and alkali denaturation method	25–40	1–14 days after birth + 4.5–38 weeks after birth	SaO2	Hemoximeter OSM2	Nellcor N-100 (Hayward, CA, USA)	NA	For FHbF < 50% and SaO2 ≤ 95%, SpO2 was overestimated.

FHbF = fraction of fetal hemoglobin, HbF = fetal hemoglobin, NA = not applicable, paO2 = partial arterial oxygen pressure, SaO2 = arterial blood oxygen saturation, SpO2 = peripheral arterial oxygen saturation measured by pulse-oximetry, tc-pO2 = transcutaneous oxygen tension, 2,3-DPG = 2,3- diphosphogylcerate.

**Table 2 children-08-00361-t002:** Hemoximetry studies (after 1992) comparing SpO2 monitoring to invasively measured blood oxygenation parameters.

Ref	1st Author, Year	Number of Patients/ HbF Blood Samples	Blood Sample Type	HbF Measurement Method	Gestation Distribution (Weeks)	Time of Sample Collection and Non-Invasive Monitoring	Blood Oxygenation Parameters	Blood Gas Analyzer/Hemoximeter	Pulse Oximeter (Company Name)	Additional Bedside Oxygenation Monitoring Device (Company Name)	Relevant Results
[6].	Rajadurai, 1992	22/64	Arterial	Visible absorption spectroscopy (hemoximeter)	25–36	1 h–73 days after birth	Functional SaO2 *	ABL30 Analyzer/ Hemoximeter OSM3	Nellcor N-100 (Hayward, CA, USA)	NA	Pulse oximeter saturations were unaffected by FHbF values which ranged from 0 to 100%.
[29].	Shiao, 2005	20/210	Arterial and venous	Visible absorption spectroscopy (hemoximeter)	24–34	First 5 days after birth	paO2, SaO2, SvO2, HbO2	Hemoximeter OSM3	Nellcor NPB 290 (Pleasanton, CA, USA)	NA	Bias of SpO2 vs HbO2 was +1.6% (2SD 5.6) and SpO2 vs SaO2 −0.6% (2SD 5.9). There was no statistical analysis of HbF contribution to the bias.
[30].	Shiao, 2006	39/188	Arterial and venous	Visible absorption spectroscopy (hemoximeter) + HPLC	25–38	First 5 days after birth	paO2, SaO2, SvO2, HbO2	Hemoximeter OSM3	Nellcor NPB 290 (Tyco Healthcare, Mansfield, MA, USA)	NA	Lower HbF levels after the transfusion resulted in lower SpO2 for the same paO2 range of 50–75 mmHg. There was no statistical analysis of HbF contribution to the SpO2-SaO2 bias.
[31].	Shiao, 2007	78/771	Arterial and venous	Visible absorption spectroscopy (hemoximeter)	25–38	First 5 days after birth (every 6–8 h)	paO2, SaO2, HbO2	Hemoximeter OSM3	Nellcor (NPB 290, Pleasanton, CA, USA)	SaO2m, SvO2m *** Oximetric 3-wavelength monitors (Abbott, Chicago, IL, USA)	Bias of SpO2 vs HbO2 in arterial blood samples was 2.5% (SD 3.1). There was no statistical analysis of HbF contribution to the SpO2-SaO2 bias.
[34].	Nitzan, 2018	14/28	Arterial	Visible absorption spectroscopy (hemoximeter)	24–33	Within 12 h before and after the blood transfusion (first 5 days after birth)	paO2, SaO2	ABL 90 FLEX	Nellcor (Covidien-Medtronic, Mansfield, MA, USA)	NA	HbF declined significantly after transfusion and FiO2 increased by > 12% to keep SpO2 within the same range.

FHbF = fraction of fetal hemoglobin, HbF = fetal hemoglobin, HbO2 = oxyhemoglobin, HPLC = high performance liquid chromatography, NA = not applicable, paO2 = partial arterial oxygen pressure, SaO2 = arterial blood oxygen saturation, SvO2 = venous blood oxygen saturation, SpO2 = peripheral arterial oxygen saturation measured by pulse-oximetry, SD = standard deviation. *Functional SaO2 = (HbO2/ 100− HbCO- HbMet) × 100, **SO2 = SaO2 and SvO2, *** SaO2m = arterial blood oxygen saturation monitoring, SvO2m = venous blood oxygen saturation monitoring (measured through umbilical catheters by using Oximetric 3 monitors of 3-wavelength technology).

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
