# Peer review of "Accuracy of Pulse Oximetry in the Presence of Fetal Hemoglobin—A Systematic Review"

_children, 2021, doi:10.3390/children8050361_

Round 1

Reviewer 1 Report

Maybe I have to apologise for finding the paper a bit confusing:

In the abstract (page 1, line 13-15) the authors state that: "The presence of fetal hemoglobin (HbF) in the blood of neonates leads to a leftward shift in the oxyhemoglobin dissociation curve and might, therefore, affect SpO2 monitors, which are calibrated using adult blood."

As for the leftward shift of the oxyhemoglobin dissociation curve (ODC), there is a broad consensus. However, while this shift indicates a different relation of oxygen saturation and oxygen pressure it should not in itself affect the SpO2 monitor.

One would rather expect the SpO2 reading to be influenced by the presence of HbF if its light absorption properties were signficantly different from HbA, which - according to many authors they are not - at least in the wavelength region commonly used in pulse oximetry (see Ref below, including Ref 26 of the manuscript).

Based on the analysis of ten papers, the authors conclude that "on the results of the majority of the included studies, the presence of HbF in neonatal blood can lead to an overestimation of SpO2" (page 9, line 3-4). However, they state "the results of these four studies remain rather inconclusive" (page 9, line 18) and, concerning "The fifth study of the period before 1992", "the reported results are difficult to interpret as well".

The main results of the remaining 5 papers, three of which are authored by the same person (Shiao) are summarized in Table 2 (page 7).

According to Ref 6 SpO2 was not affected by FHbF.

From the data given for Ref 6, 29-31 and 34 on cannot draw clear conclusions on to what extent HbF was supposed to influence SpO2 readings. Is it possible that these papers basically confirm that the presence of HbF is associated with a leftward shift of the ODC?

In summary, there is not sufficent data to support the conclusion, that "high fractions of HbF could have caused an overestimation of SpO2" (page 11, line 104).

Nevertheless, it seems reasonable taking into account the influence of Hbf on the OCC when discussing optimal oxygen delivery for preterm infants immediately after birth.

Harris AP, Sendak MJ, Donham RT, Thomas M, Duncan D.

Absorption characteristics of human fetal hemoglobin at wavelengths used in pulse oximetry.

(Ref 26) Zijlstra WG, Buursma A, Meeuwsen-van der Roest WP.

Absorption spectra of human fetal and adult oxyhemoglobin, de-oxyhemoglobin, carboxyhemoglobin, and methemoglobin.

Clin Chem. 1991 Sep;37(9):1633-8. PMID: 1716537.

Nijland R, Jongsma HW, Nijhuis JG, Oeseburg B, Zijlstra WG.

Notes on the apparent discordance of pulse oximetry and multi-wavelength haemoglobin photometry.

Acta Anaesthesiol Scand Suppl. 1995;107:49-52. doi: 10.1111/j.1399-6576.1995.tb04330.x. PMID: 8599297.

Whyte RK, Jangaard KA, Dooley KC.

From oxygen content to pulse oximetry: completing the picture in the newborn.

Acta Anaesthesiol Scand Suppl. 1995;107:95-100. doi: 10.1111/j.1399-6576.1995.tb04341.x. PMID: 8599308.

Mendelson Y, Kent JC.

Variations in optical absorption spectra of adult and fetal hemoglobins and its effect on pulse oximetry.

IEEE Trans Biomed Eng. 1989 Aug;36(8):844-8. doi: 10.1109/10.30810. PMID: 2474489.

Reviewer 2 Report

In this review manuscript, the authors summarize existing evidence on the influence of HbF on pulse oximetry readings. The subject is of great importance as many therapeutic decisions on respiratory and oxygen support in the NICU are based on the monitoring of arterial oxygen saturation by pulse oximetry (SpO2). They searched for studies that compared arterial oxygen saturations (SaO2) from neonatal blood with SpO2 values with the inclusion of HbF in the analyses. The manuscript is comprehensibly written. However, there are a few points that need to be addressed:

Major points:

The authors found 10 studies that met the inclusion criteria. They state that 6 of those found that HbF had an effect on non-invasive SpO2 monitoring and 4 showed inconclusive results or no significant effect of HbF on SpO2 values. I do not agree with this interpretation.

Only one of the 6 studies demonstrates an influence of HbF on the difference between SpO2 and SaO2 values (Jennis 1987, HbF >50% generated a 2.8% to 3.6% error (underestimation) in SpO2 reading). However, in this study SpO2 values were lower than SaO2 values. This might be explained by the majority of oxygen saturation values >95%.

Two studies might show an influence of HbF on the bias between SpO2 and SaO2: Nitzan 2018 and Praud 1989. The primary outcome of Nitzan et al. 2018 was to investigate the effect of blood transfusion on saturation targeting in neonates.

HbF declined significantly after transfusion and FiO2 increased by >12% to keep SpO2 within the same range. This effect is explained by the shift of the ODC of HbF and HbA. The difference between SpO2 and SaO2 values before transfusion was 1.9% ± 2.2% and after transfusion 0.2% ± 1.1%. However, it is not clear from this study whether this is solely due to the decrease in HbF or whether the change in other parameters as pH or methemoglobin as a result of the transfusion might have an influence on the bias as well. Furthermore, no statistics are presented to show whether the differences were significant. Praud et al. (1989) demonstrated as stated correctly in Table 1 of the review manuscript, that for HbF <50% and SaO2 ≤95%, SpO2 was overestimated. However, this statement is incomplete and therefore misleading in its message. Praud et al. also found that the difference between SpO2 and SaO2 decreased with higher HbF (HbF <50% +4.2% ±2.4%, HbF >50% +0.9% ±1.8%). Therefore, HbF cannot be the only factor causing this difference.

Finally, the studies by Shiao et al. (2005, 2006 and 2007) did not analyze the influence of HbF on the relation between SpO2 and SaO2 but between oxygen saturation parameters primarily derived from blood analyses. In 2005 Shiao et al. found that without adjusting for the effects of HbF mean SO2 measurements were elevated for 5% (±1.38%). Meant is here the adjustment for HbF (neonatal / adult mode) of the analyzer (Hemoximeter OSM3). Without the adjustment the bias in SaO2 is 5% - not the comparison to SpO2. However, they found differences in SpO2 vs. HbO2 of 1.6% (2SD 5.6%) and SpO2 vs. SaO2 of -0.6% (2SD 5.9%) in HbF mode. Furthermore, they stated that HbF levels had significant effects on the differences between HbO2 and SaO2 (total blood hemoximeter) measurements. With each percent increase of HbF, there was a 0.02% increase of the difference between the measurements. But they did not find an influence of HbF on any other comparison / difference. In their 2006 study the primary goal was the comparison of different HbF measurements. They found as stated correctly in Table 2 of the review manuscript that the bias of the hemoximeter was 23% (±9.1) against the HPLC. Meant here is the bias between different HbF measurements. Furthermore, they found that lower HbF levels after a blood transfusion resulted in lower SpO2 for the same paO2 range of 50-75 mmHg. This is due to the shift in the ODC. In one figure (Figure 2) a difference between SpO2 and SaO2 is perceivable: before transfusion SpO2 is about 1% higher than SaO2, after transfusion SpO2 is about 1% higher below an oxygen saturation of 95% and about 1 to 2% lower than SaO2 above an oxygen saturation of 95%. However, no statistical evaluation is presented and it remains unclear whether these differences are significant and whether they are due to the change in HbF levels. Finally, the 2007 study of Shiao et al. presents confusing statements. They found that the mean difference between SpO2 and oxyhemoglobin in arterial blood samples was +2.5% (SD 3.1). However, any influence of HbF is only reported on blood derived oxygen saturation parameters. In the results section they state for the comparison of blood oxygen saturation and oxyhemoglobin: “Each 1% increase in fetal hemoglobin was associated with a 0.027% decrease (P=.04) in the mean difference between oxygen saturation and oxyhemoglobin (with an intercept of 5.26, P < .001), with blood transfusion factor contributing to a 0.45% increase in the mean difference (P < .001). This result indicated that the greater the percentage of fetal hemoglobin, the smaller the difference between the blood oxygen saturation and blood oxyhemoglobin measurements.” However, in the discussion section they suddenly draw the opposite conclusion: “…the effects of fetal hemoglobin on the differences between oxygen saturation and oxyhemoglobin measurements changed along the oxyhemoglobin dissociation curves, with greater differences when fetal hemoglobin levels were higher.” This is greatly confusing and hampers the interpretation of this study in that matter.  

Finally, all of the other studies found differences between the measurements of SpO2 and SaO2 within the range of -2.5 to 2.9%. None of these studies found an influence of HbF on these differences. Wimberley et al. furthermore report on the influence of multiple factors on the SaO2 - paO2 relation in preterm neonates, however, they do not descripe how this conclusion was drawn and whether any differences were higher or lower with increasing HbF.

In summary, the results from the presented studies remain inconclusive at best and the conclusion the authors draw needs to be more cautiously phrased. An increase in HbF changes the relation of SpO2 to paO2 which is physiologically explained by the shift in the ODC, but does not say anything about the accuracy of pulse oximeter readings. It is important to be aware of this shift in order to avoid oxygen toxicity. As analyzers today usually correct for the presence of HbF the measured SaO2 likely reflects the paO2 of the neonates’ blood correctly. However, whether HbF is solely responsible for the reported differences between SpO2 as measured by pulse oximeter and SaO2 determined in blood is not conclusively proven. There well might be an influence of HbF on SpO2 readings causing an overestimation of SpO2, especially as the calibration of the pulse oximeters is based on adult blood samples. This one has to bear in mind when treating a neonate, especially for the lower SpO2 limits in preterm neonates. However, further studies are needed to support this statement.

Minor points:

Introduction:

Consider shortening the second paragraph of the introduction.

Page 1, lines 45-46: consider rephrasing to: … the ratio of absorbance at these wavelengths is calculated and calibrated against direct measurements of arterial oxygen saturation from blood samples (SaO2). For this purpose, blood samples are taken…

Page 2, line 63: should be: … from the blood of the mother to the fetus via the placenta…

Figure 1: the abbreviations for pO2 and SO2 are already explained within the figure, HbA does not appear in the figure, those explanations can be dropped from the legend

Material and Methods:

Page 3, line 117: it should probably read: …on January 24th, 2021.

Results:

Page 4, line 129: The reference 28 does not fit.

Page 4, lines 129-132: All studies also determined blood oxygenation parameters (as this was part of the inclusion criteria).

Page 4, line 146: it must read: …whereby all affected SaO2 and paO2 in neonates… (as correctly stated in Table 1)

Table 1 and 2: the last column is confusing as results of the primary outcomes of the studies are presented which not always are congruent with the reviews purpose. Consider rephrasing the last column in primary outcome and adding another column with results relevant for this review.

Discussion:

Page 9, line 12: it should read: …which could be attributed to HbF…

Page 10, line 90: it should read: …the studies did not report…

Round 2

Reviewer 2 Report

I do not have further comments/ recommendations